# Vulnerability-Aware Parameter-Efficient Fine-Tuning for Enhanced Adversarial Robustness

## Abstract

Pre-trained foundation models (PTMs) that undergo standard pre-training can be efficiently finetuned for downstream tasks using parameter-efficient fine-tuning (PEFT) methods. However, these models remain highly vulnerable to adversarial perturbations. Existing studies often distribute PEFT parameters uniformly across layers, which overlooks the varying importance of each layer. In this work, we systematically analyze the adversarial robustness of PEFT strategies and introduce a novel *vulnerability score*, a computationally efficient gradient-based measure that identifies which layers and components are most susceptible to adversarial attacks. Guided by this score, we design robustness-aware PEFT methods: *LoRA High*, which concentrates parameters in the most vulnerable layers, and *LoRA+Adapter*, which assigns LoRA to the attention component and adapters to the feed-forward component. Extensive adversarial-training experiments across four real-world image classification datasets show that these targeted PEFT designs consistently outperform vanilla PEFT methods. Post-adversarial finetuning analysis with pruning-style attribution score confirms that strategically protecting vulnerable parts of the backbone is key to robustness in PEFT.

## 1 Introduction

Deep Learning models have achieved great success in diverse application domains, including computer vision He et al. (2016), natural language processing Devlin (2018), and robotics Lenz et al. (2015). Recently, with the emergence of foundation models, parameter-efficient fine-tuning has gained popularity because of the cost-effective way to adapt the large model to downstream tasks, achieving good performance. Parameter-efficient fine-tuning methods (PEFT) introduce additional lightweight parameters to the networks that can be trained by keeping the backbone fixed. There have been various PEFT methods introduced: Prompt Lester et al. (2021); Jia et al. (2022), LoRA Hu et al. (2022), Adapter Houlsby et al. (2019); Pfeiffer et al. (2020b;a), Bias Cai et al. (2020); Zaken et al. (2021), and Linear Probe Chen et al. (2021). Although these methods can be used to achieve good performance on downstream tasks, adapting the model using these methods also makes the model more susceptible to adversarial attacks.

Understanding the robustness of PTM-based models is essential to developing reliable models that can be deployed in safety-critical environments. Starting from the study of adversarial robustness for Convolutional Neural Networks (CNNs) Cui et al. (2021); Wang et al. (2023); Liu et al. (2023), earlier works have also studied the adversarial robustness for fine-tuning foundation models Mahmood et al. (2021). However, these methods mainly focus on adversarial pretraining or full fine-tuning of the pre-trained models, which is computationally expensive. Thus, recent works have started studying the robustness of using PEFT methods to fine-tune the pretrained foundation models Xu et al. (2024); Lv et al. (2024); Yuan et al. (2025); Hua et al. (2024); Li et al. (2025). Yuan et al. (2025) found that introducing additional normalization in LoRA and using LoRA in different parts of the transformer block improves the adversarial robustness compared to using LoRA only in multi-head attention (MHA) and multilayer perceptron (MLP) layers. Hua et al. (2024) studied the robustness of different PEFT methods, including LoRA, Prompt, Adapter, and Linear Probe, and observed that robust fine-tuning is better than standard fine-tuning using PEFT; however, full fine-tuning is often more robust than PEFT. Despite this progress, existing studies largely distribute PEFT parameters

uniformly across the transformer layers, without identifying *which parts of the frozen backbone are most vulnerable* to adversarial perturbations. Such parts refer to either different layers or components within each layer of the network. Moreover, past works assume access to PTMs that are pre-trained to be robust, which are expensive and not always available. We argue that not every part of the network is equally important for adversarial robustness, and the strength of the PEFT parameters in different parts of the network should be allocated based on the importance of that part for adversarial robustness.

We hypothesize that standard PTMs without robust pre-training can freely rely on shortcut but predictive patterns in a small region of the input image. Thus, an attacker can easily break the model by exploiting such patterns to make a small but visually undetectable change in the input. However, we hypothesize that not all parts of the model are vulnerable to such shortcut patterns. Therefore, we seek to identify the vulnerable parts in the network and adjust the PEFT capacity accordingly to protect such vulnerable parts. To identify the vulnerable parts of the network, we introduce a novel score-based metric: *vulnerability score*, associated with each parameter of the network. For each parameter, this score is obtained by evaluating the average gradient norm across the adversarial samples. This is further verified by the vulnerability score of the clean model for the CIFAR10 dataset before adversarial finetuning in Figure 1, where layers 0, 1, and 4 are the 3 most vulnerable layers as they rank the highest in vulnerability score; similarly, layers 9, 10, and 11 are the 3 least vulnerable layers. This provides evidence that not all layers are equally vulnerable. Additionally, the computation cost of the vulnerability score is cheaper as it is computed before performing adversarial training and using a model finetuned with linear PEFT on clean samples.

We analyze the vulnerability at two levels: (i) **Layer Level:** for a given layer, the vulnerability scores of the selected parameters are summed to get the vulnerability score of that layer. The layers with the largest vulnerability score should be given more importance. Guided by this, we propose *LoRA High* PEFT, which concentrates LoRA parameters on top-$k$ vulnerable layers. In contrast, we further propose and compare with *LoRA Low* PEFT, which focuses LoRA parameters on the least-$k$ vulnerable layers, keeping the total number of parameters the same as *LoRA High*. The results verify that *LoRA High* consistently outperforms *LoRA Low* in adversarial robustness evaluation. (ii) **Component Level:** Within each transformer layer (block), we find that on average, the MHA component tends to rank higher in vulnerability score compared to the MLP component. However, the vulnerability score of MLP is also non-negligible and is competitive with the score of MHA for some layers. To cover both components, we propose a simple *LoRA+Adapter* PEFT design that assigns LoRA to the MHA component and adapter to the MLP component, yielding complementary robustness gains.

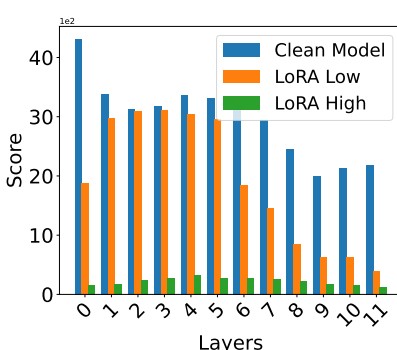

Figure 1: Layer-level vulnerability score before and after robust finetuning.

After adversarial finetuning with the proposed PEFT methods, we observe that the vulnerability scores of the targeted part of the backbone drop the most compared to adversarial finetuning with vanilla PEFT methods, indicating the importance of correcting the right vulnerable parts. This is supported by the evidence in Figure 1 that LoRA High reduces the vulnerability scores across the layers better than LoRA Low after robust finetuning. To further explain why these placements are effective, we also make use of pruning-style neuron attribution (e.g., SNIP Lee et al. (2018)) scores. We find that the PEFT methods that concentrate importance on fewer, more vulnerable parts can tolerate higher pruning ratios with smaller robustness degradation than vanilla PEFT methods.

Our key contributions are summarized as follows:

- We identify the most vulnerable parts of the network by introducing a novel and computationally efficient *vulnerability score* which is an adversarial gradient measure computed from a model finetuned using the linear PEFT on clean samples without any adversarial pretraining.

- We provide a layer-level and component-level analysis of vulnerability, which led us to design the LoRA High and LoRA+Adapter PEFT method that focuses on allocating more PEFT parameters for the most vulnerable parts of the model.

- We carry out extensive adversarial training experiments across CIFAR10, CIFAR100, CUB, and ImgNetR datasets, where the proposed PEFT methods achieve strong robustness, demonstrating their effectiveness.

- To the best of our knowledge, our work is the first to leverage the neuron attribute score, such as the SNIP score, to explain why certain PEFT methods in ViTs show better adversarial robustness than others

## 2 RELATED WORKS

**PEFT Methods** Adapter Houlsby et al. (2019); Pfeiffer et al. (2020b;a) introduce lightweight down-projection and up-projection parameters to either introduce them in sequential or parallel manner to the MLP layer of the transformer block. Prompt Lester et al. (2021); Jia et al. (2022) introduce trainable parameter on the input side and append it into the patch embedding of the input before passing to the transformer blocks. LoRA Hu et al. (2022) introduces trainable lightweight projection matrices in the MHA layer of the transformer blocks. Bias Cai et al. (2020); Zaken et al. (2021) unfreezes the bias parameters of the pre-trained backbone. Linear Probe Chen et al. (2021) only adapts the classifier head by keeping the rest of the weights fixed.

**Adversarial Robustness** Earlier works focus on adversarial robustness when performing full fine-tuning or robust pre-training. Mahmood et al. (2021) studies the robustness of Vision Transformers (ViT) and checks whether adversarial samples generated from CNNs or ViT transfer to each other. Cui et al. (2021) works on adversarial robustness of CNNs to prevent the drop of clean accuracy by leveraging the logits from the clean model. Wang et al. (2023) combines multiple types of adversarial attacks during robustness training to improve the robustness of CNNs. Liu et al. (2023) focuses on transferring the robustness of a pretrained robust model to downstream tasks while maintaining the learned robustness of the pre-trained model. Xu et al. (2024) found that gradients for standard and adversarial objectives are conflicting, which leads to poor adversarial robustness and introduces a method that combines different training objectives in the loss function. They both focus on adversarially pre-trained ResNet models. Lv et al. (2024) performs an ensemble of different LoRA parameters corresponding to different adversarial training tasks. Yuan et al. (2025) introduces a new normalization method on top of the original LoRA and uses the modified LoRA in different layers of the transformer block in ViT. Hua et al. (2024) focuses on adversarially finetuning a robust model using different PEFT techniques and introduces a novel initialization method that further improves the robustness. Li et al. (2025) studies the trade-off between clean accuracy and robustness and whether we can maintain the trade-off after performing finetuning using different PEFT methods. Apart from starting with non-robust and standard PTMs, our method differs from the existing closely related works in the following ways: 1) Instead of allocating the PEFT parameters uniformly across different parts of the network, we introduce a vulnerability score metric to identify the most vulnerable parts of the network and concentrate the PEFT parameters more on the vulnerable parts. 2) Our method is the first to provide post-adversarial finetuning analysis of why certain PEFT methods are more robust with the help of neuron attribution scores, such as the SNIP score, along with our proposed vulnerability score.

**Neuron Attribution** Neuron attribution scores are used to determine the neurons and parameters most responsible for the performance of the model. Lee et al. (2018) introduces a simple way to compute such a score (SNIP score) that is used to prune parameters in a model, making it sparse and efficient to train. This score focused on the global loss function and the specific weight to prune. Sun et al. (2023) introduces the WANDA score, which focuses on the local region of the model by using the input activation and weights to prune parameters in a given layer. Wei et al. (2024) makes use of these scores to explain the robustness of Large Language Models towards safety-alignment. However, to the best of our knowledge, our work is the first to explain the adversarial robustness of different PEFT methods using SNIP scores for vision tasks using ViT.

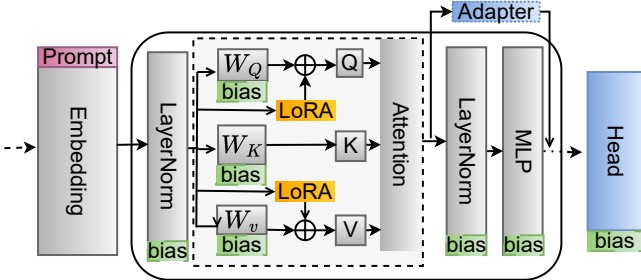

Figure 2: Figure shows the location of different PEFT techniques across the ViT architecture. The Prompt, LoRA, Bias, and Adapter are highlighted in color.

## 3 PRELIMINARIES

**Problem setup**  Consider a classification task where the input data $\mathcal{D} = \{(x_n, y_n)\}_{n=1}^N$ contains $N$ number of training samples. Here $x_n \in \mathcal{X}$ and $y_n \in \mathcal{Y}$ are the individual inputs and the corresponding label from the input set $\mathcal{X}$ and label set $\mathcal{Y}$. Let $\theta$ correspond to the model we want to be adversarially robust.

We aim to study the robustness of existing pre-trained models with a focus on the impact of the choice of PEFT on the effectiveness of adversarial robustness. We want to fine-tune the foundation model using different PEFT methods such that robustness is learned during fine-tuning. We also call this robust fine-tuning. To this end, we consider a popular transformer architecture and PEFT-based robust fine-tuning as shown in Figure 2. We consider the most popular PEFT methods, including Prompt Jia et al. (2022), Bias Zaken et al. (2021), Adapter Zaken et al. (2021), and LoRA Hu et al. (2022). Prompt introduces additional learnable prompt parameters that are concatenated with the input embedding; Bias unfreezes all the bias parameters of the pre-trained model; Adapter introduces additional learnable adapter modules; and LoRA introduces learnable low-rank matrices. During adversarial robustness training, the backbone is frozen and only the PEFT parameters are trained.

**Generating Adversarial Samples**  We consider norm-bounded adversaries that craft examples within an $\ell_p$ ball of radius $\varepsilon$ around an input $\mathbf{x}$ that maximizes the loss:

$$\mathbf{x}_a = \mathbf{x} + \delta : \delta = \underset{\|\delta\|_p \leq \varepsilon}{\arg\max} \ \mathcal{L}(\theta; \mathbf{x} + \delta, \mathbf{y}), \tag{1}$$

Here, $\mathcal{L}$ is cross-entropy loss, $\mathbf{x}_a$ is the adversarial sample, and $\ell_p$ is commonly $\ell_\infty$ or $\ell_2$. We mainly use adversarial attacks based on Projected Gradient Descent (PGD)  Madry (2017), which approximates equation 1 with an iterative objective where the attack starts from a clean sample $\mathbf{x}_{t=0} = \mathbf{x}$ for $T$ total iterations and at each iteration $t$:

$$\mathbf{x}_{t+1} \leftarrow \Pi_{\mathcal{B}_p(\mathbf{x}, \varepsilon)}\big(\mathbf{x}_t + \alpha \operatorname{step}_p(\nabla_\mathbf{x} \mathcal{L}(\theta; \mathbf{x}_t, \mathbf{y}))\big), \tag{2}$$

Where $\operatorname{step}_\infty(\mathbf{g}) = \operatorname{sign}(\mathbf{g})$, $\operatorname{step}_2(\mathbf{g}) = \mathbf{g}/\|\mathbf{g}\|_2$, $\alpha$ is the step size, and $\Pi_{\mathcal{B}_\epsilon(\mathbf{x})}(\cdot)$ denotes the projection operator that ensures the perturbed example remains within an $\epsilon$-ball $\mathcal{B}_\epsilon(\mathbf{x})$ around the original input $\mathbf{x}$ under a specified norm.

**Adversarial Training with PEFT**  Pre-trained foundation models (PTMs) are trained to achieve good generalization performance. Explicitly training the PTMs to be adversarial robust is computationally expensive, and most PTMs are trained without considering adversarial robustness. Such training is likely to guide the PTMs to learn spurious correlations that can ingrain adversarial vulnerabilities in the model. The robustness of a model $\theta$ can be improved with the help of adversarial training, where the model is trained on adversarial samples Madry (2017); Goodfellow et al. (2014). However, due to long training time, high memory requirements, and expensive data costs, it is not feasible to train the foundation models from scratch with the adversarial training objective. An effective adversarial PEFT technique is required that can ensure both robustness and efficiency. To this

end, we use an approach where the PEFT parameters of the model are learned using the adversarial samples. We formulate Adversarial Training Madry (2017) (AT) as :

$$\min_{\theta} \mathbb{E}_{(\mathbf{x},y)\sim\mathcal{D}}\Big[\mathcal{L}(\theta,\mathbf{x}_a,\mathbf{y})\Big] \tag{3}$$

where $\mathcal{L}(\cdot)$ is the loss function, such as cross-entropy loss, $\mathbf{x}_a$ is an adversarially attacked sample corresponding to the clean sample $\mathbf{x}$, and the PEFT parameters are updated during adversarial training.

## 4 VULNERABILITY AWARE ROBUST PEFT

As the adversarial attack exploits the gradient direction where the model is most sensitive to the predictions, we hypothesize that the success of adversarial attacks hinges on vulnerable pathways in the network that locally amplify small, structured input changes into large loss increases. Standard PTMs, not trained for invariance to small perturbations, can rely on shortcut patterns that are predictive but not robust. The adversarial attacks can easily exploit the network paths that rely on such shortcut features. However, not all parameters of the network are responsible for amplifying the shortcut features. Thus, in the following section, we attempt to identify the vulnerable parameters in the network that amplify the shortcut features.

**Identification of Vulnerability with Vulnerability Score**   Let $\frac{\partial\mathcal{L}}{\partial\mathbf{x}}$ denote the sensitivity of loss with change in the input. We are interested in finding the vulnerability of a parameter $W$ which measures the amplification of loss sensitivity for an input $\mathbf{x}$. The vulnerability score thus can be obtained by the approximation of the following mixed partials:

$$\begin{aligned} V(W,\mathbf{x}) &= \frac{\partial}{\partial W}\frac{\partial\mathcal{L}(\theta,\mathbf{x},\mathbf{y})}{\partial\mathbf{x}} = \frac{\partial}{\partial\mathbf{x}}\frac{\partial\mathcal{L}(\theta,\mathbf{x},\mathbf{y})}{\partial W} \\ &\approx \frac{\nabla_W\mathcal{L}(\theta,\mathbf{x}+\Delta\mathbf{x},\mathbf{y}) - \nabla_W\mathcal{L}(\theta,\mathbf{x},\mathbf{y})}{\Delta\mathbf{x}} \\ &\propto \|\nabla_W\mathcal{L}(\theta,\mathbf{x}+\Delta\mathbf{x},\mathbf{y})\| \end{aligned} \tag{4}$$

Where we use the first order approximation of $f(\mathbf{x}) = \frac{\partial\mathcal{L}(\theta,\mathbf{x},\mathbf{y})}{\partial\mathbf{w}}$ and assume $|\nabla_W\mathcal{L}(\theta,\mathbf{x}+\Delta\mathbf{x},\mathbf{y})| \gg |\nabla_W\mathcal{L}(\theta,\mathbf{x},\mathbf{y})|$ as $\theta$ is optimized for clean sample $\mathbf{x}$. Let $\mathcal{D}_a$ denote adversarial examples generated from a clean set $\mathcal{D}$ by equation 2. We define the *parameter-level vulnerability* as the expected gradient norm on adversarial inputs:

$$V(W) \triangleq \mathbb{E}_{(\mathbf{x}_a,\mathbf{y})\sim\mathcal{D}_a}\Big[\|\nabla_W\mathcal{L}(\theta;\mathbf{x}_a,\mathbf{y})\|\Big]. \tag{5}$$

To compare parameters of different sizes/scales, we can use a *scale-invariant* variant: $\tilde{V}(W) \triangleq V(W)/(\|W\| + \sigma)$ where $\sigma$ is a small positive value added to prevent division by zero.

**Addressing Vulnerability**   We attempt to address the vulnerability at two levels: (1) Layer Level and (2) Component Level. For layer-level vulnerability, we identify the transformer blocks (layers) that are more vulnerable than others. For a transformer layer $l$ with disjoint blocks $W \in \mathcal{W}_l$, we define the *layer-level* vulnerability score as: $V_l \triangleq \sum_{W\in\mathcal{W}_l}\tilde{V}(W)$ where $\mathcal{W}_l$ is the set of weights for layer $l$. Thus, we choose the top-$k$ most vulnerable layers with the largest $V_l$ and modify the PEFT associated with that layer. The PEFT can be modified to concentrate more PEFT parameters toward the most vulnerable layers than other layers. For example, for LoRA, we can increase the LoRA rank and scale hyperparameter for those vulnerable layers. To demonstrate the effectiveness of our method, we propose LoRA High and LoRA Low, which increase the strength of LoRA for the most vulnerable layers and the least vulnerable layers, respectively.

For component-level vulnerability, we make an attempt to first analyze the importance of the MHA and MLP components within each transformer block (layer). For component $c \in \{\text{MHA}, \text{MLP}\}$ within layer $l$ the component level vulnerability score is given by: $V_{l,c} \triangleq \sum_{W\in\mathcal{W}_{l,c}}\tilde{V}(W)$ where

$\mathcal{W}_{l,c}$ is the set of weights for component $c$ within layer $l$. Figure 3 shows that MHA components, on average, are more vulnerable than MLP layers. However, we noticed that MLP components are also vulnerable, and the vulnerability is not too far compared to MHA layers. Thus, we realize that just focusing on the MHA component does not solve the robustness problem, and focusing on the MLP component is also important. To this end, we propose LoRA+Adapter PEFT, where the LoRA PEFT takes care of the MHA component and Adapter PEFT takes care of the MLP component.

**Connection with low-level features learning**   Previous work Raghu et al. (2021) explores the similarity of ViTs with CNNs and found that in CNNs, the early layers are more focused on detecting the low-level features such as colors and textures, while the later layers focus on high-level features, such as objects. However, in ViT, the low-level features can be learned throughout the ViT layers, where learning low-level features early on is also important. As the adversarial attacks make small changes, we hypothesize that they mostly focus on the low-level features. Therefore, the vulnerability score can help detect which layers focus on these low-level features the most, as those layers could be most vulnerable. Further, our results also show that, on average, earlier layers are more vulnerable than later layers; however, the middle layers are also vulnerable, which supports the claim of Raghu et al. (2021) that low-level feature learning is spread across transformer layers in ViT with slightly higher concentration in the earlier layers.

**Post-adversarial finetuning Analysis**   After performing adversarial finetuning, we check the vulnerability score of the parameters that we tried to fix using the vulnerability score. If the vulnerability score is reduced after PEFT, it proves the effectiveness of that PEFT method. Further, we make use of the SNIP score Lee et al. (2018); Wei et al. (2024) to verify the effectiveness of the PEFT methods, as robust PEFT methods should be able to retain robustness when a larger number of parameters are pruned. The SNIP score is computed after the robust finetuning of the model using PEFT and is given by: $I(W) = \mathbb{E}_{x \sim \mathcal{D}'} |W \nabla_W \mathcal{L}(x)|$. Here, $W \in \mathbb{R}^{d_o \times d_i}$ is the weight matrix of any linear layer where $d_o$ and $d_i$ are the shape of the output layer and input layer, respectively. $\mathcal{D}'$ could be either clean samples or adversarially perturbed samples. The shape of $I(W)$ is the same as $W$, such that $I(W)$ contains the element-wise importance score of each parameter in $W$. $\mathcal{L}$ refers to the cross-entropy loss. If $\mathcal{D}'$ contains clean samples, $I(W)$ represents the importance score of each parameter w.r.t clean samples; such that parameters with the largest values of $I(W)$ in $W$ are the parameters most important to clean samples. Similarly, we can obtain the parameters most important to adversarial samples, where such parameters can also be referred to as the most responsible for adversarial robustness.

## 5  EXPERIMENTS

We first describe the datasets and the baselines used in the experiments. We then study the impact of PEFT choice on both the generalization and robustness after the adversarial training. Finally, we carry out ablation studies on a different adversarial attack. Our source code can be found at this link: https://anonymous.4open.science/r/padv-8EF5/

**Experiment Setup**   We consider 4 benchmark datasets of CIFAR10, CIFAR100 Krizhevsky et al. (2009), ImgNetR Hendrycks et al. (2021), and CUB Wah et al. (2011). We select the ViT-B16 transformer architecture as our base model. We compare the robustness of 4 vanilla PEFT techniques: Prompt, LoRA, Adapter, and Bias, and 3 augmented PEFT techniques: LoRA High, LoRA Low, and LoRA+Adapter. For the main result, we use a cross-entropy loss-based standard PGD attack where the attack strength ($\epsilon/255$) of the adversarial attack is controlled by $\epsilon$. For robust finetuning, the attack strength, step size, and number of steps are fixed to 3/255, 0.01, and 2, respectively. For evaluation, the step size and number of steps are fixed to 0.004 and 20, respectively. For Prompt, we append a 768-dimensional token of length 5 as a learnable prompt to the input. For the adapter, we use the bottleneck of size 16 with output dimensions of 768. For LoRA, we use a rank of 4. For LoRA High (or LoRA Low), we select the top-$k$ (or least-$k$) most vulnerable layers and increase the rank to 64 while keeping the rank of the remaining layers as 4. For the result reported in our paper, we select $k = 3$. For LoRA+Adapter, we initialize LoRA with the parameters obtained from robust finetuning using vanilla LoRA.

Table 1: Adversarial Robustness Comparison of different PEFT Methods After Adversarial Training.

| Method | Trainable Parameters | $\epsilon$ | CIFAR10 | CIFAR100 | CUB | ImgNetR |
|---|---|---|---|---|---|---|
| Prompt | 11520 | 0 | 49.30 | 31.26 | 69.85 | 46.00 |
| | | 1 | 40.88 | 24.66 | 46.09 | 31.85 |
| | | 2 | 32.94 | 18.64 | 22.94 | 20.80 |
| | | 3 | 25.19 | 13.24 | 08.56 | 12.63 |
| | | 5 | 12.50 | 05.49 | 01.06 | 04.77 |
| Adapter | 304320 | 0 | 36.58 | 48.46 | 81.13 | **74.25** |
| | | 1 | 40.24 | 34.19 | 71.29 | 60.43 |
| | | 2 | 27.02 | 21.30 | 54.41 | 45.93 |
| | | 3 | 13.97 | 13.79 | 38.08 | 32.87 |
| | | 5 | 01.94 | 03.94 | 14.12 | 15.20 |
| Bias | 102912 | 0 | 86.01 | 69.67 | 82.40 | 69.20 |
| | | 1 | 78.49 | 60.20 | 71.79 | 59.05 |
| | | 2 | 69.44 | 49.43 | 57.76 | 49.35 |
| | | 3 | 58.13 | 39.04 | 42.88 | 39.67 |
| | | 5 | 33.87 | 20.05 | 19.80 | 23.07 |
| LoRA | 147456 | 0 | 89.10 | 36.72 | 82.49 | 73.50 |
| | | 1 | 83.45 | 28.89 | 72.39 | **62.20** |
| | | 2 | 75.47 | 34.99 | 58.31 | 49.95 |
| | | 3 | 65.84 | 31.50 | 44.49 | 37.50 |
| | | 5 | 42.22 | 08.57 | 21.59 | 19.35 |
| **LoRA+Adapter** | 451776 | 0 | **90.50** | **75.27** | 80.70 | 69.80 |
| | | 1 | **85.14** | **67.04** | 70.48 | 60.73 |
| | | 2 | **78.14** | 56.96 | 56.79 | 50.83 |
| | | 3 | **68.84** | 46.28 | 43.94 | 41.72 |
| | | 5 | **45.91** | 26.19 | 22.52 | 24.62 |
| **LoRA Low** | 700416 | 0 | 90.07 | 74.56 | 71.93 | 70.58 |
| | | 1 | 75.54 | 65.87 | **73.62** | 61.03 |
| | | 2 | 56.94 | 55.71 | **59.20** | 50.68 |
| | | 3 | 40.69 | 44.57 | 44.87 | 41.17 |
| | | 5 | 27.29 | 25.24 | 21.92 | 24.25 |
| **LoRA High** | 700416 | 0 | 90.41 | 74.53 | **82.53** | 70.53 |
| | | 1 | 84.76 | 66.55 | 73.16 | 62.02 |
| | | 2 | 77.29 | **57.12** | 59.07 | **52.55** |
| | | 3 | 67.88 | **46.49** | **46.18** | **42.92** |
| | | 5 | 44.87 | **26.86** | **23.58** | **26.40** |

**Robustness Evaluation** Table 1 shows the robustness of PEFT methods across different attack strengths ($\epsilon$) for the PGD attack. $\epsilon = 0$ refers to the generalization on clean samples. Among the PEFT methods, Prompt has the least robustness as it introduces trainable parameters only in the input layer and the classifier head. Results show that among LoRA, Bias, and Adapter, LoRA is generally able to achieve better robustness. The robustness of LoRA may be attributed to the topology of the introduced PEFT parameters. LoRA adds a trainable parameter to the MHA layer, which seems to be important than the adapter that only adds a trainable parameter to the MLP layer of the transformer block. The MHA layer may be more important because adversarial attacks make the model prone to using spurious shortcut features, and thus, it may be more important for the model to shift the attention to the correct features. We further observe that the robustness of Bias is also competitive when compared with the robustness of LoRA. This may be because Bias can enable trainable parameters across multiple layers of a transformer block. This evidence supports that introducing trainable parameters in multiple layers is better for robustness.

These vanilla PEFT methods distribute the PEFT parameters uniformly across the transformer layers. However, from our theoretical analysis, not all parts of the network are equally responsible for the vulnerability towards adversarial attacks. To identify the most important parts of the network, we first evaluate the component-level vulnerability score of MHA and MLP components as shown in Figure 3. We should note that the layer-level vulnerability score can be obtained by summing the score of MHA and MLP components for each layer. Using the vulnerability score, we compare LoRA High and LoRA Low PEFT in Table 1. We observe that LoRA High consistently shows better robustness than LoRA Low. This verifies our analysis that concentrating the PEFT parameters in the vulnerable parts yields better robustness compared to placing them in the least vulnerable parts for

the same parameter size. Further, Figure 3 shows that the MLP component is also vulnerable with non-negligible vulnerability, which motivates us to combine LoRA and Adapter. We observe that in Table 1, LoRA+Adapter also has better robustness compared to other vanilla PEFT methods in most cases. In the case of the CUB dataset, LoRA+Adapter and LoRA High can only beat LoRA Low in the strongest attack strength ($\epsilon = 3, 5$), whereas the robustness is slightly lower than LoRA Low for lower $\epsilon$ values. This may be because of the nature of the dataset. For instance, the CUB dataset only contains 5,994 training samples of all bird images, in which LoRA Low may already be performing best, leaving less room for improvement for weaker attacks.

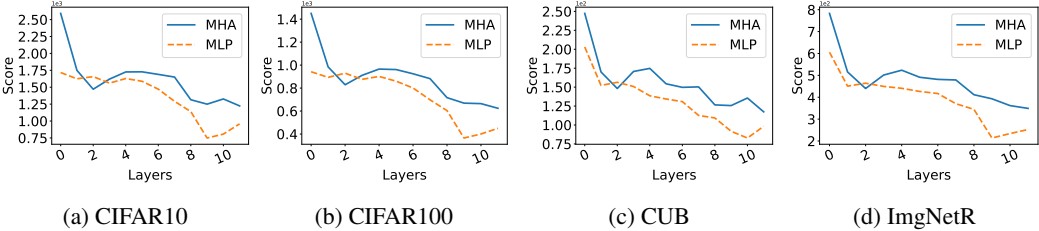

(a) CIFAR10          (b) CIFAR100          (c) CUB          (d) ImgNetR

Figure 3: The component-level vulnerability scores $V_{l,c}$ across the layers before robust PEFT.

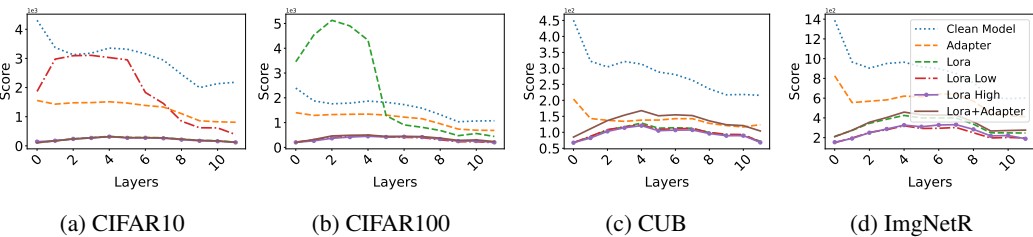

(a) CIFAR10          (b) CIFAR100          (c) CUB          (d) ImgNetR

Figure 4: The layer-level vulnerability scores $V_l$ across the transformer layers for different methods.

**Reduction of Vulnerability After Adversarial Fine-tuning**    As we propose PEFT methods that focus on the most vulnerable parts, we are interested in seeing if the vulnerability of those parts is reduced after adversarial finetuning. Figure 4 shows the layer-level vulnerability score across the transformer layers. After adversarial finetuning, the vulnerability score drops compared to the clean model. We further observe that LoRA+Adapter and LoRA High reduce the vulnerability score of these components the most compared to other PEFT methods. This provides evidence that selecting the most important layers is important in order to fix the vulnerability and thus improve the adversarial robustness.

**Pruning Unimportant Parameters**

Motivated by Wei et al. (2024), we also perform a pruning analysis where the parameters least important to robustness are set to zero. We evaluate the SNIP score for adversarial samples with attack strength $\epsilon = 3$ under the PGD attack. The SNIP score assigns higher scores to parameters important for adversarial robustness. Given a pruning ratio $r$ and any linear weight $W \in \mathbb{R}^{d_o \times d_i}$ with SNIP scores $I(W)$, for each row (across output neurons) of $W$, we replace the value of $r\%$ of parameters having the least SNIP scores with zero. This is performed for all trainable linear parameters for the given PEFT method, excluding the

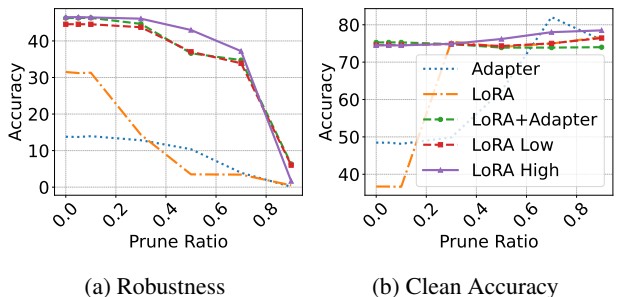

(a) Robustness          (b) Clean Accuracy

Figure 5: Adversarial robustness and clean accuracy on CIFAR100.

classifier head. To compute the SNIP score, we use the adversarial samples generated using $\epsilon = 3$ and also evaluate on $\epsilon = 3$ where other settings are the same as in the main Table 1.

Figure 5 shows that a good fraction of unimportant parameters can be pruned without the loss of robustness or clean accuracy. In Figure 5a, the adversarial robustness for LoRA High and LoRA+Adapter can still be maintained even after pruning around 30% of unimportant parameters. It shows that a large fraction of parameters from LoRA High and LoRA+Adapter can be pruned before the adversarial robustness fails. Further, LoRA High is consistently more robust to pruning compared to other PEFT methods. This also verifies that LoRA High introduces a larger number of parameters useful for adversarial robustness compared to other PEFT methods. In Figure 5b, as the pruning ratio increases, the clean accuracy is also maintained for LoRA High and LoRA+Adapter, whereas for LoRA and Adapter, the clean accuracy improves only when more PEFT parameters are pruned. We suspect this is because, as the PEFT parameters are gradually removed, the pre-trained original backbone has more precedence in determining the final output. Since the original backbone has not gone through adversarial training, it is more suited towards clean samples, which will thus improve the clean accuracy by sacrificing the adversarial robustness. In contrast, as fewer parameters are pruned, the robustness improves while sacrificing the clean accuracy.

**Ablation Studies** In this section, we perform the robustness analysis of LoRA, Adapter, LoRA+Adapter, LoRA Low, and LoRA High for the auto-PGD Croce & Hein (2020b) attack. For the adversarial training, we train for 5 steps for CIFAR10 and 2 steps for ImgNetR with $\epsilon = 3$. For the evaluation, the number of steps is increased to 20 with different values of $\epsilon$. The result in Table 2 shows that auto-PGD is generally stronger than the PGD attack in Table 1. Nevertheless, we observe a similar pattern that LoRA is generally better than Adapter in robustness, and the LoRA High and LoRA+Adapter further improve the robustness. In the case of ImgNetR, the combination is slightly lower than LoRA in weaker attack strength, but as the attack strength increases, the robustness is improved for the combination. In the case of CIFAR10 dataset, LoRA+Adapter, LoRA High and LoRA Low have performance closer to each other, suggesting that there may be less room for improvement for LoRA High. However, in the ImgNetR dataset, it is clear that LoRA High is better than LoRA Low for adversarial robustness, supporting the importance of concentrating LoRA parameters for the most vulnerable layers.

Table 2: Adversarial Robustness Comparison of different PEFT Methods After Adversarial Training for Auto-PGD attack.

| Dataset | $\epsilon$ | Adapter | LoRA | LoRA+Adapter | LoRA Low | LoRA High |
|---------|---|---------|------|--------------|----------|-----------|
| CIFAR10 | 0 | **97.99** | 92.63 | 93.34 | 93.61 | 93.24 |
|         | 1 | 64.37 | 86.09 | **87.55** | 87.63 | 87.17 |
|         | 2 | 42.61 | 76.85 | **78.66** | 78.51 | 78.1 |
|         | 3 | 25.39 | 63.54 | **65.77** | 65.34 | 65.15 |
|         | 5 | 04.20 | 32.05 | **34.74** | 33.38 | 33.68 |
| ImgNetR | 0 | 73.07 | 72.58 | 70.93 | **77.55** | 73.55 |
|         | 1 | 59.47 | 61.03 | 60.53 | 56.32 | **62.03** |
|         | 2 | 45.57 | 48.75 | 48.52 | 41.45 | **50.28** |
|         | 3 | 31.77 | 36.23 | 36.90 | 30.13 | **38.48** |
|         | 5 | 13.52 | 17.17 | 18.33 | 14.23 | **19.55** |

## 6 CONCLUSION

We studied the adversarial robustness while robust fine-tuning PTMs using PEFT and showed that robustness depends not only on the number of trainable parameters but also on where they are placed. We introduced a novel score-based metric that identifies layers and components most susceptible to adversarial perturbations. Guided by this score, we designed robustness-aware PEFT strategies: *LoRA High*, which allocates capacity to top-$k$ vulnerable layers, and *LoRA+Adapter*, which complements MHA with LoRA and MLP with adapters. These methods consistently improve adversarial robustness over vanilla PEFT across real-world datasets. Our results highlight that robustness in PEFT improves from strategically protecting the most vulnerable parts of the backbone. Moreover, post-adversarial finetuning analysis SNIP score confirms that targeting vulnerable parts enables the model to tolerate pruning with smaller robustness degradation.

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

# A    APPENDIX

The Appendix is organized as follows: first, we provide a table that summarizes the symbols used in the paper. Next, we include the algorithm for adversarial training and calculating the vulnerability score. Finally, we include additional training details and additional experiments.

Table 3: Summary of the Symbols and their Definitions

| Symbol | Definition |
|---|---|
| $\mathcal{D}$ | Set of clean dataset |
| $N$ | Size of dataset $\mathcal{D}$ |
| $x$ | Single entry of dataset $\mathcal{D}$ |
| $y$ | The label for the data point $x$ |
| $\theta$ | The parameters of the model, including the PEFT parameters and the backbone |
| $x_a$ | The adversarially attacked sample corresponding to the clean sample $x$. |
| $W$ | Any linear weight |
| $d_o$ | Number of output neurons for a given linear weight $W$ |
| $d_i$ | Number of input neurons for a given linear weight $W$ |
| $I(W)$ | SNIP scores corresponding to weight $W$ |
| $V(W)$ | Vulnerability score corresponding to weight $W$ |
| $\epsilon$ | Attack strength |
| $\alpha$ | Attack step size |
| $\mathcal{L}$ | Cross entropy loss |
| $L(\cdot)$ | Cost function for attack |
| $\mathcal{B}_\epsilon$ | $\epsilon$ ball around $x$ |

## A.1    SUMMARY OF SYMBOLS

Table 3 summarizes the symbols used throughout the paper.

## A.2    ALGORITHM FOR CALCULATION OF VULNERABILITY SCORE

Algorithm 1 provides the algorithm to compute the vulnerability score and to perform the adversarial robust training.

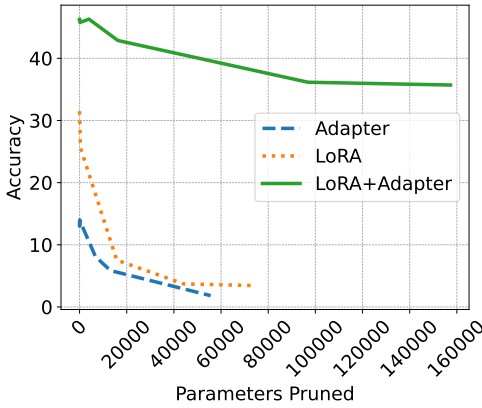

Figure 6: Pruning parameters important to robustness for the CIFAR100 dataset.

---

**Algorithm 1** Parameter Efficient Adversarial Training and Vulnerability Score Calculation

---
**Step 1: Adversarial Training**

**Input**: Dataset $\mathcal{D}$
**Output**: Learned Model $f^\theta$

1: Initialize Model Parameter $\theta \leftarrow \theta_0$
2: Generate Adversarial Samples $\mathcal{D}_a$ using Equation 2.
3: Train $\theta$ on $\mathcal{D}_a$ using Cross-Entropy loss
4: **return** $\theta$

**Step 2: Vulnerability Score Calculation**

**Input**: Model $\theta_0$, Clean samples $\mathcal{D}$ chosen weight $W$
**Output**: Vulnerability score $V(W)$

1: Finetune a Linear PEFT model while freezing the backbone to obtain $\theta$ on clean samples $\mathcal{D}$.
2: Using the obtained model $\theta$, attack the clean samples $\mathcal{D}$ to compute the set of adversarial samples $\mathcal{D}_a$
3: Using the adversarial samples $\mathcal{D}_a$ compute the gradient of cross-entropy loss wrt the chosen backbone parameter: $\nabla_W \mathcal{L}(\mathbf{x})$.
4: Compute the vulnerability score $V(W)$ using equation 5.
5: **return** $V(W)$

---

### A.3 ADDITIONAL EXPERIMENT RESULTS

Here we first provide the additional experiment details, followed by additional results.

**Additional Experiment Details:** For the adversarial training, we only train on the adversarial samples for 30 epochs with a batch size of 48. We use the SGD optimizer and cosine scheduler with an initial learning rate of 0.03, weight decay of 0.0005, and a minimum learning rate of 0. For the combination of LoRA and Adapter, we initialize the LoRA parameters with LoRA parameters obtained after robust finetuning of LoRA by itself. This is because LoRA generally has stronger robustness compared to Adapter, and initializing with a strong position helps to further improve the robustness. We also fix the seed to 1993 across NumPy, PyTorch, and Python.

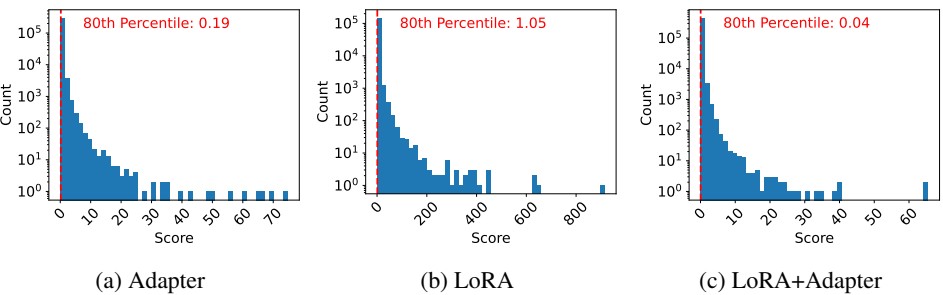

(a) Adapter      (b) LoRA      (c) LoRA+Adapter

Figure 7: Distribution of SNIP score for trainable parameters for CIFAR100 dataset.

**Distribution of SNIP score** We visualize the overall SNIP score distribution for different PEFT methods in Figure 7. We observe that the distribution is very sparse, such that there is a lower percentage of parameters with higher scores. For example, in Figure 7a, 80% of the adapter parameters have scores less than 0.19 while the maximum score is 74.82. This suggests that there is a very small proportion of parameters that help with the adversarial robustness.

**Pruning Parameters Important for Robustness:** In Figure 6, we prune important parameters for robustness, but exclude the top 10% important parameters for clean accuracy to compute the set difference. The robustness of the Adapter falls off quickly, followed by LoRA as more parameters are pruned. The LoRA+Adapter does not drop significantly because there are more important pa-

rameters for robustness, as well as some important parameters for robustness that also overlap with parameters important to clean performance, which is protected by the set difference.

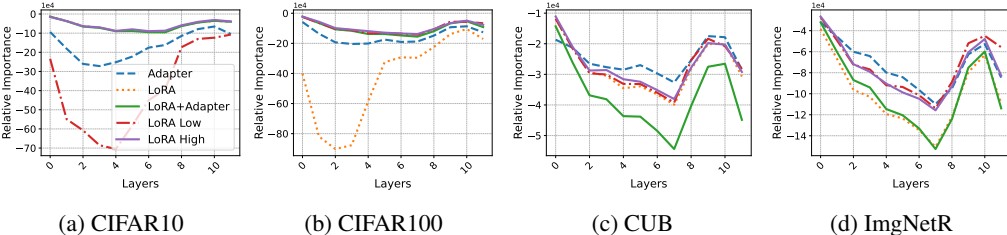

|            (a) CIFAR10            |            (b) CIFAR100           |               (c) CUB              |             (d) ImgNetR            |

Figure 8: Relative Importance: difference between the SNIP (importance) score of PEFT parameters and backbone parameters.

**Relative Importance of PEFT Parameters:** Figure 8 shows the difference between the SNIP score of PEFT parameters and backbone parameters across the transformer layers. It shows that the importance of PEFT parameters with respect to the backbone parameters increased for LoRA High and LoRA+Adapter compared to other PEFT methods. This is consistently true for the CIFAR10 and CIFAR100 dataset. However, for the CUB and ImgNetR datasets, the pattern is consistent only for early layers. This may be because the SNIP score is not directly connected to the robustness but rather the importance of the parameters. It shows that for these datasets, the backbone parameters are more important in the later layers compared to earlier layers.

Table 4: Adversarial Robustness against different types of Attack for CIFAR10 dataset

| Model        | PGD   | APGD  | FAB   | APGDT |
|--------------|-------|-------|-------|-------|
| Adapter      | 13.97 | 25.39 | 31.67 | 13.34 |
| LoRA         | 65.84 | 63.54 | 62.3  | 61.42 |
| LoRA+Adapter | **68.84** | **65.77** | **64.52** | **63.72** |

**Adversarial Robustness Against Different Attacks** Table 4 shows the adversarial robustness of Adapter, LoRA, and LoRA+Adapter across PGD Madry (2017), APGD, APGDT Croce & Hein (2020b), and FAB Croce & Hein (2020a) for CIFAR10 dataset. The pattern remains consistent that LoRA has better robustness than the adapter, and LoRA+Adapter is better than LoRA and Adapter individually. For PGD, the adversarial training and evaluation are the same as the experiment in Table 1. For adversarial training against APGD, FAB, and APGDT, we use the same setting to generate the adversarial samples as in Table 2, where the APGD attack with attack strength $\epsilon = 3$ and number of steps of 5 is used. After training on APGD, the model is evaluated on APGD, FAB, and APGDT attacks with the number of steps of 20.

