# OpenReview forum: "Vulnerability-Aware Parameter-Efficient Fine-Tuning for Enhanced Adversarial Robustness"
_ICLR.cc/2026/Conference — ICLR 2026 Conference Withdrawn Submission_

### Official Review · Reviewer_Yp1q · 2025-10-25

**Soundness:** 3
**Presentation:** 3
**Contribution:** 2
**Rating:** 6
**Confidence:** 3

**Summary:**

- Introduces a cheap, adversarial-gradient–based vulnerability score to rank layers/components by attack sensitivity.

- Uses the score to allocate PEFT capacity: LoRA-High (focus LoRA on top-k vulnerable layers) and LoRA+Adapter (LoRA in MHA, adapter in MLP) to cover both components.

- Across CIFAR-10/100, CUB, and ImageNet-R, these targeted placements generally improve robust accuracy over vanilla PEFT

- Provides interpretability/diagnostics: targeted PEFT lowers vulnerability scores the most and tolerates higher pruning (via SNIP-style analysis).

**Strengths:**

- Simple, actionable idea: a vulnerability score that guides where to place PEFT capacity without changing the base architecture or training recipe.

- Consistent empirical improvements across datasets and attacks, showing clear gains over uniform LoRA/Adapter baselines.

- Interpretable diagnostics: layer/component vulnerability analysis and pruning behavior help explain why targeted placement works.

- Easy to reproduce and integrate: uses standard ViT backbones and common PEFT modules, making adoption practical.

**Weaknesses:**

- Backbone clarity: the exact ViT setup isn’t clearly positioned as a modern “foundation” model. That makes it harder to judge how generally applicable the findings are.

- Limited foundation-model evidence: there’s no demonstration on true FM-class backbones (e.g., large ViT/CLIP/SigLIP checkpoints). Without that, it’s unclear whether the vulnerability-guided placement still helps with strong pretraining.

- Parameter-budget fairness: different trainable-parameter counts. It’s difficult to isolate placement benefits from capacity/initialization effects without strictly matched baselines.

- Scalability & generality: only one backbone family and small/mid-scale datasets. No tests on larger ViTs, other architectures (ConvNeXt/Swin), or tasks beyond classification.

- What about general performance? The paper doesn’t convincingly show the impact on non-robust utility. Two risks to probe: (1) clean accuracy regressions (catastrophic forgetting) and (2) degraded transfer/zero-shot. They should report clean top-1, zero-shot on ImageNet-1k (if using CLIP-like FMs), linear-probe/VTAB-style transfers, and maybe retrieval/captioning if the base is multimodal; ideally showing either no loss or a clear robustness–utility trade-off curve.

**Questions:**

I have no questions - I am willing to increase the score if the above concerns are addressed

---

### Official Review · Reviewer_juXX · 2025-10-28

**Soundness:** 2
**Presentation:** 3
**Contribution:** 1
**Rating:** 2
**Confidence:** 5

**Summary:**

This paper proposes vulnerability score, a gradient-based measure that identifies if a layer is susceprible to adversarial attacks. Based on such score, the paper designs robustness-aware PEFT method that focuses the finetuning on the most vulnerable layers. The score-enhanced PEFT shows consistent improvement over vanilla PEFT methods.

**Strengths:**

1. The paper explores PEFT method for adversarial training, which is an important topic given the currently increasing model size and presisting robustness issues in deep leanring models
2. The idea of measuring the difference in layer's sensitivity to adversarial attack is well-motivated
3. The method description is clear and easy to follow. Overall presentation is good.

**Weaknesses:**

1. Novelty-wise, the paper misses an important line of related work, layer-selective adversarial training [1,2,3], in the discussion. This hinders the novelty claim of the proposed vulnerability score. Existing work already explores different variants of layer robustness measurements to select only sensitive layers for finetuinng and combines the layer selective tuning with PEFT methods like LoRA and DoRA [3]. Discussion of the novelty and superiority of the proposed vulnerability score against existing methods is needed to justify its novelty and significance.
2. The design and hyperparameter choices of the proposed method seems arbitrary and not well-justified. For example, in line 76 the vulnerability score of a layer is the sum of its parameters' scores. However, since the score is defined as a gradient norm, it needs justification why sum is a good choice, compared to a more natural choice of concatnating the gradients then do a norm together. The choice of top-K selection, LoRA ranks, adversarial training hyperparameters are all dictated without justification or ablation study.
3. Only PGD is used as the base adversarial training method in the experiments. However, more advanced adversarial training method exists that can improve the robustness-accuracy tradeoff against PGD baseline. More experiments are needed to show the generalizability of the proposed vulnerability score across different models, base adversarial training methods, and hyperparameter choices.
4. As shown in Table 1, the effectiveness of the proposed method is not consistent on CUB and ImgNetR. Higher robustness is achieved at the cost of a lower clean accuracy, which may not be a desired behavior. It is hard to justify the effectiveness of the tradeoff with the current evaluation


[1] RiFT, ICCV 2023, https://arxiv.org/abs/2308.02533

[2] CLAT, ICML 2025, https://openreview.net/forum?id=mMwuRYIPFm

[3] SAFER, ICCV 2025, https://arxiv.org/abs/2501.01529

**Questions:**

1. The result in Fig. 1 shows that the vulnerability score will drop for all layers under LoRA-high finetuning, even if most of the layers are not finetuned. Why is this the case? Does this mean the vulnerability score is not a consistent measure of a single layer's property, but acts more like a measurement of the whole model robustness?
2. How does the proposed method generalize across different adversarial training methods and hyperparameters?
3. What is the impact of training hyperparameters, like LoRA rank and layer selection k, on the overall results of the proposed method?

---

### Official Review · Reviewer_LCKe · 2025-10-29

**Soundness:** 1
**Presentation:** 2
**Contribution:** 2
**Rating:** 2
**Confidence:** 4

**Summary:**

This paper studies how to make parameter-efficient fine-tuning (PEFT) methods more robust against adversarial attacks. The authors find that different layers in pre-trained models have different levels of vulnerability, which standard PEFT methods ignore. They propose a gradient-based vulnerability score to identify the most fragile layers and components, then design two robustness-aware PEFT methods. Experiments on four image classification datasets show that these targeted methods improve both robustness and overall performance compared to standard PEFT.

**Strengths:**

The paper is easy to follow.

**Weaknesses:**

The main concern is that I don’t really see the benefit of using vulnerability-aware PEFT over standard PEFT. For vulnerability-aware PEFT, you first need to identify the vulnerable layers, which can differ across model architectures or sizes. While there might be some general trends (e.g., vulnerable layers often appearing in early or middle stages), determining them still requires non-trivial computation. The only potential advantage seems to be a reduction in trainable parameters. However, in Table 1, the number of trainable parameters is actually higher than in LoRA due to the larger rank, which eliminates this advantage. Moreover, both the clean accuracy and robustness are worse than those achieved by LoRA + Adapter, which is a combination of standard PEFT methods. Therefore, I have no reason to choose vulnerability-aware PEFT over standard PEFT.

**Questions:**

What is the advantage to use vulnerability-aware PEFT over standard PEFT?

---

### Official Review · Reviewer_dW6w · 2025-11-01

**Soundness:** 2
**Presentation:** 3
**Contribution:** 2
**Rating:** 2
**Confidence:** 5

**Summary:**

This paper investigates the adversarial robustness of Parameter Efficient Fine-Tuning (PEFT) methods for Vision transformers (ViT), specifically, Low Rank Adaptations (LoRA), while providing a focus on providing more of a focus on more important layers for robustness, through analyzing the gradient norm with respect to the weights. They provide mostly sufficient experiments highlighting the effectiveness of their methods and showcasing the performance of other PEFT methods.

**Strengths:**

- The problem setting is relatively novel and well-motivated, and the methodology is sound

**Weaknesses:**

- Experiments suffer from imbalanced scenarios for different competing methods (see questions)
- Certain derivations are mathematically not sound/not well-motivated (see questions)
- Novelty of the method itself is relatively low, and there are no significant theoretical contributions. The experiments are not extensive enough to make up for this.

**Questions:**

- Line 86: “we find that on average, the MHA component tends to rank higher in vulnerability score compared to the MLP component. However, the vulnerability score of MLP is also non-negligible and is competitive with the score of MHA for some layers.” I believe the evidence of this is in figure 3, the authors should reference the figure here. Additionally, I think it would be good to add the post-robust PEFT scores as well to match that of fig 1 or fig 4.

- Related works: Existing works [1] have discussed the importance of individual layers during Adversarial PEFT (in this case, prompt-tuning) and provide supporting evidence about prompting the early and middle layers. I believe this is relevant and should at least be discussed in related works.

- Line 241: Considering $x$ is a vector and therefore so is $\Delta x$ , line 2 of Eq 4. **is not well defined**. The authors should either change the notation or remove line 2 entirely. More importantly, the motivation behind choosing the mixed partial derivative as the score function is unclear since, once again, it is not a scalar. Since the metric is heuristic, experimental evidence such as the one requested in the following two paragraphs would be a better motivation in my view for using the gradient norm as a score.

- Line 248: $f(x) = \frac{\partial \mathcal{L}}{\partial w}$ is not a correct first order approximation, unless one is considering $f(w)$ as a function of w, in which case the partial derivative would be an approximation of $f(w + \Delta w) - f(w)$, not $f(w)$.  Regardless, this approximation is not required for the argument that follows, as it is reasonable to assume the gradient would be zero/near zero, assuming the trained model parameters are a local minima for the clean samples.

- Additionally, regarding line 248 and line 255, the assumption that the norm of the gradients of the perturbed sample is higher is easily verifiable empirically, and I think the authors should provide evidence of this. Furthermore, why not use the norm of the perturbed samples' gradient minus or divided by the norm of the clean samples' gradient rather than divide by the norm of W? I believe this is additionally informative in the case of comparing the scores of MHA and MLP modules, since the norm of one depends on the other as the gradients are computed through backpropagation.

- Table 1: To ensure a fair comparison, I believe the number of parameters for vanilla LoRA and LoRA + Adapter should be increased to match that of LoRA high/low. The same goes for the other methods, but to a lower degree due to the different shapes of parameters. Regardless, as it stands, each method is tuning a widely different number of parameters (within a two orders of magnitude difference), which makes the comparison unfair. **This should be addressed if the results are to be credible.**

- Figure 5: Does not include scores for prompt tuning, should be added.

As of now, I remain skeptical of the paper, however, should all of the above be addressed, I would consider raising my score to marginally accept.

[1] Eskandar M, Imtiaz T, Wang Z, Dy J. ADAPT to Robustify Prompt Tuning Vision Transformers. arXiv preprint arXiv:2403.13196. 2024 Mar 19.

---

### Note · Authors · 2025-11-14

I have read and agree with the venue's withdrawal policy on behalf of myself and my co-authors.